# Peptides and Dendrimers: How to Combat Viral and Bacterial Infections

**DOI:** 10.3390/pharmaceutics13010101

**Published:** 2021-01-14

**Authors:** Annarita Falanga, Valentina Del Genio, Stefania Galdiero

**Affiliations:** 1Department of Agricultural Science, University of Naples “Federico II”, Via dell’Università 100, 80100 Portici, Italy; 2Department of Pharmacy, University of Naples “Federico II”, Via Domenico Montesano 49, 80131 Naples, Italy; valentina.delgenio@unina.it

**Keywords:** peptide, dendrimer, antimicrobial

## Abstract

The alarming growth of antimicrobial resistance and recent viral pandemic events have enhanced the need for novel approaches through innovative agents that are mainly able to attach to the external layers of bacteria and viruses, causing permanent damage. Antimicrobial molecules are potent broad-spectrum agents with a high potential as novel therapeutics. In this context, antimicrobial peptides, cell penetrating peptides, and antiviral peptides play a major role, and have been suggested as promising solutions. Furthermore, dendrimers are to be considered as suitable macromolecules for the development of advanced nanosystems that are able to complement the typical properties of dendrimers with those of peptides. This review focuses on the description of nanoplatforms constructed with peptides and dendrimers, and their applications.

## 1. Introduction

The human body is colonized by various bacteria and viruses that can be pathogenic or nonpathogenic. The microbiota consists of bacteria, viruses, and eukaryotes that cooperate with a host’s immune system, living in equilibrium; therefore, the microbiota can perform its functions in an effective and synchronized way [1]. Nonetheless, bacterial and viral infections can seriously compromise our health. Humans are continuously in contact with potential pathogenic bacteria and viruses; luckily, the human immune system is trained against their invasions, and not all infections need medical intervention; in fact, some bacterial and viral infections can trigger mild and moderate diseases, while others cause the death of millions of people. The recent and ongoing pandemic outbreak of SARS-CoV2 (COVID-19) that began in early 2020 further supports the necessity to face these global challenges through the advancement of our ability to handle infectious diseases [2]. Although bacterial and viral infections may trigger similar symptoms, such as weakness, fever, and a state of general inflammation, the infections are caused by very dissimilar microbes; in fact, bacteria are rather complex organisms able to live alone, while viruses need to colonize/invade a cell/living organism to reproduce.

Bacteria are prokaryotic, unicellular microorganisms with a rigid cell wall, which allows them to preserve a definite shape and size (0.2–30 μm). Pathogenic bacteria are responsible for numerous infections, and have an important impact on human health. Gram-negative bacteria are enveloped by a thin peptidoglycan layer comprising an inner cytoplasmic membrane and an outer membrane. The outer membrane, which is absent in Gram-positive bacteria, slows down molecular diffusion and limits antibacterial diffusion, and is likely responsible for the higher resistance that Gram-negative bacteria have to common antiseptics and disinfectants compared to Gram-positive bacteria.

Viruses are obligate parasites of the host cell. The infection begins with the attachment of the virion to the host cell and the interaction between the proteins exposed on the viral surface and one or several receptors on the host cell membrane. Entry into the host cells is the first step of the viral life cycle, which is followed by translation, replication, assembly, and egress [3,4]. After viral attacks, an extensive host-cell reorganization takes place, due to the presence of viral proteins in an appropriate sub-cellular compartment. In this way, the viral infection is described in different phases of viral activity (attack, penetration, and replication), and each phase may provide useful targets in the design of antiviral strategies.

Antimicrobial resistance, which is a natural event widely diffused in the world, represents one of the biggest global health challenges of our time, and renders the advancement of therapeutic strategies urgent [5]. Increased resistance produces increased morbidity and mortality due to infectious diseases worldwide, and fighting this trend requires collaborative efforts among research sectors (Figure 1).

For bacterial infections, despite the role of antibiotics that have contributed to a significant reduction in diseases caused by bacteria, the development of resistance has complicated the scenario. In particular, elements contributing to the occurrence of resistance are the indiscriminate overuse, inappropriate use, inadequate dosing, and poor adherence to treatment guidelines for antibiotics, such as prolonged therapies or prophylaxis, and use in agriculture and animal production [6]. Moreover, the diffusion of antimicrobial resistance is favored by poor infection control, inadequate sanitary settings, and inappropriate food handling. The World Health Organization (WHO), together with the Food and Agriculture Organization of the United Nations (FAO) and the World Organization for Animal Health (OIE), supports the ‘One Health’ approach for best practices aimed at avoiding the emergence and diffusion of antibiotic resistance, and favoring optimal use of antibiotics in both humans and animals [7]. The increasing trend toward multidrug resistance has diminished the options for effective therapeutic drugs for bacterial infections even more. The principal mechanisms exploited by bacteria to develop multiple drug resistance are: (i) changes in cellular permeability to prevent entry of antibiotics into cells, (ii) changes in antibiotic targets to prevent their action, and (iii) exploitation of efflux pumps to push antibiotics out of the cell interior [8].

Antiviral drug resistance is also a matter of great clinical importance; a huge number of viral infections are treated with antiviral drugs, which generally target mechanisms of viral penetration and replication. A robust treatment will impair a viral genome from successfully replicating, but if the treatment is not as effective and some genomes still replicate, then selective pressure may lead to rapid adaptation toward resistance [9]. Antiviral resistance has been described for almost all approved drugs with mechanisms of resistance, varying among viruses and among specific drug regimens. Therefore, there continues to be a need for novel drugs to treat the ever-expanding number of patients infected with drug-resistant viruses [10].

Limiting overuse and misuse of antimicrobials may limit and/or reduce the resistance issue, with consequent postponing of the increase in antimicrobial resistance. Alternative and preventive measures that will replace the use of antimicrobials in the future are being explored. In this view, one particularly appealing strategy could be the use of polymers, which possess antimicrobial activity, combined with other antimicrobial molecules such as peptides, to develop nanotherapeutics based on peptides [11]. Antimicrobial peptides hold great promise due to their ability to produce physical disruption of the microbial membranes, which could likely reduce the incidence of resistance, as it would be much more difficult for the bacteria to undertake membrane repair. Polymers with innate antimicrobial activities and that are apt to conjugate to other antimicrobial molecules represent a versatile and appealing strategy; in fact, the activity of the conjugated molecules could be enhanced, and those molecules could be selected according to the infectious pathogen. Thus, the choice of the polymer is a crucial issue. Research in polymer development has opened the possibility to design and synthetize new materials with antibacterial activity correlated to their chemical structures and/or able to improve the potency of other molecules. Chemists have developed a variety of polymers with cationic and amphiphilic structures that proved to be effective as antimicrobials. A fine-tuning of amphiphilicity and charge is key to antimicrobial activity and selectivity. A series of pioneering works by Kuroda showed how even a non-peptide system with a random sequence can selectively kill bacteria when hydrophobic and cationic moieties are inserted into the structure and finely tuned [12,13,14]. The charged group and its density are also two key features; for example, guanidinium groups have enhanced activity compared to amines, likely because they can generate bidentate hydrogen bonds with lipids [15]. In fact, polymers with a higher charge density show lower hemolytic toxicity, but similar antimicrobial activity compared to less-charged polymers [16]. Another key challenge is the biodegradation of polymers in order to avoid or minimize unwanted side effects caused by prolonged retention of the materials inside the human body or in the environment [17]. Among polymers, dendrimers deserve great attention; they have been widely used due to their high multivalence, ease of synthesis and functionalization with peptides, and biological activity [18]. While dendrimers (Figure 2) have garnered great interest in their applications in drug and gene delivery, their function as antimicrobial agents is still in its infancy, and represents a great challenge to fighting antimicrobial resistance. Recent literature shows that dendrimers may be effective against biofilms, may act in synergy with antibiotics, and may act against multi-drug-resistant bacteria and against viruses.

There is a plethora of reviews on dendrimers and peptides, but most of these deal separately with the drug-delivery potential of the nanocarriers and the antimicrobial activity of peptides. Here, we review the therapeutic potential of dendritic polymers covalently conjugated to peptides in the development of antimicrobials. This review is meant to provide a brief summary of the principal concepts concerning the development of viral and antibacterial peptides for the functionalization of dendrimers in order to promote further research in the field and further work on the development of dendrimer-based antimicrobial compounds.

## 2. Novel Approaches

### 2.1. The Use of Peptides

The fight against antimicrobial resistance has enhanced interest in peptide-based approaches. Peptides are multifunctional and modular; most of them possess important biological functions, and they may be exploited to create avant-garde nanotechnologies [19]. Compared to traditional drugs, peptides present several advantages: high biocompatibility, selectivity for targets, low toxicity, and easy elimination from the body. In spite of this, linear peptides containing an L amino acid have a low half-life, their oral bioavailability is hampered by the intrinsic enzymatic action, and their polarity and molecular weight may limit intestinal permeability [20]. These limits can be overcome through the design of peptidomimetics to facilitate the absorption and distribution of the drug, which can circumvent the enzymatic activity [21], allowing researchers to exploit their enormous therapeutic potential.

Peptidomimetics play an important role, since they can form the basis of new therapeutic strategies, and due to their necessary chemical modifications, can guarantee specificity and therapeutic success. By imitating typical secondary structures such as β-turn, γ-turn, α-helix, or β-strand, the affinity and selectivity for the target can be improved [22]. Several further modifications can be performed, such as: introduction of D and/or non-coded amino acids, cyclization, glycosylation, lipidation, PEGylation, or addition of other functional groups. The introduction of D amino acids and/or non-coded amino acids can significantly reduce protease activities [23]. Hydrophobicity/hydrophilicity can be modulated by combining fatty acids or cholesterol, or by adding charged residues [24]. Covalent binding of hydrophilic polymers, including polyethylene glycol (PEG), zwitterionic polymers, and others, produces better solubility and stability, and a greater half-life in vivo [25,26,27,28,29]. Cyclization also is often associated with improved stability and activity [30]. Modifications can be applied to enhance cellular internalization and delivery to the target to perform on-demand release only at specific sites with limited toxic effects [31]. Possible vectors are liposomes [32]; nanosystems based on gold, silica, or iron oxide nanoparticles; and dendrimers, all of which share the potential to be used as multivalent systems for theranostic applications (both diagnostic and therapeutic) [33].

Antimicrobial peptides (AMPs), which are short peptides that are considered the first line of defense against microbial invasions, are found in all forms of life, from bacteria to plants, fish, amphibians, insects, and mammals [34,35,36]. They are highly diverse in sequence, length, and structure, but they share a common mechanism of action that mainly involves the disruption of the integrity and function of the bacterial membrane. Most AMPs selectively kill bacteria with minimized toxicity towards eukaryotic cells; this selectivity correlates to the main differences between their cell membranes. In fact, the external layer of bacteria is negatively charged, while that of mammalian cell membranes is zwitterionic; therefore, AMPs, being positively charged, preferentially absorb to the bacterial cell surfaces through electrostatic attraction. This initial interaction is followed by the insertion of the hydrophobic domains into the lipid bilayer to disrupt the membrane, according to several models reported in literature (for a comprehensive review, readers can refer to Kang et al. [37]). Some AMPs do not cause the damage of the membrane, but cross the bilayer and target important functions of bacterial metabolism; an example includes indolicidin, which interferes with processes deputed to the formation of the cytoplasmic membrane. AMPs can be classified according to their structure: α helix, β sheet, or random coil [38]. They display a broad range of biological activities against bacteria, fungi, parasites, insects, viruses, and cancer cells. The amino-acid composition, the overall hydrophobicity, and the global charge of the molecule are essential structural features that determine their biological activity. Likewise, it is not a single parameter, but rather the contribution of several of these chemical features, that determines the biological activity. Moreover, there is not a general rule about the ideal charge and hydrophobicity of the peptide to maximize activity; there is a subtle balance that is correlated to selectivity. Some specific amino acids are highly conserved: lysines and arginines as charged residues; and alanines, valines, phenylalanines and leucines as hydrophobic residues. It is widely accepted that an excessive amount of positive charges can negatively influence activity affecting the structure of the peptide. Nonetheless, hydrophobicity also needs to be tuned to avoid toxicity to eukaryotic cells. Amphipathic conformations are critical to the interaction with the bacterial membrane, and render them less toxic for other cell types. Most AMPs are characterized by a change in conformation upon contact with the lipid membrane surface, with concomitant formation of the amphiphilic structure; moreover, self-association and multimerization upon interaction with membranes enhances their biological activity. Several models of interaction between AMPs and bacterial membranes have been hypothesized (Figure 3A). For instance, defensins show a wide antibacterial activity against several bacteria [39,40,41]. Defensins have been the object of many studies by our group to determine the smallest sequence with activity and to design molecules with enhanced activity and reduced synthetic challenges [30,39,42]. Recently, we published the design of a cyclic mini-defensin with simplified synthetic protocols and bearing the same activity as the parent peptide [42].

Another key issue is the development of biofilms and antimicrobial resistance. In particular, the biofilm matrix protects the bacteria from variations in bulk-phase conditions; thus, antibiotics kill planktonic bacteria released from the biofilm, but fail to destroy the biofilm. AMPs have recently been exploited to deal with biofilms, and they showed enhanced antibiofilm activities compared to conventional antibiotics [43,44]. We recently reported on the antibacterial activity of WMR, an analogue of myxinidin, a marine peptide derived from the epidermal mucus of hagfish [45,46]. Peptide self-assembled fibers functionalized on their external surface, with WMR showed to inhibit *Pseudomonas aeruginosa* (Gram-negative bacteria) and *Candida albicans* (pathogenic fungus) biofilm formation and eradication [47].

Differently from antibiotics, AMPs have a membrane activity that renders it very difficult and expensive for the bacteria to develop drug resistance [48]; nonetheless, some mechanisms of resistance have been reported [49].

While all AMPs possess antibacterial activity, only a few have revealed antiviral properties that target diverse steps of the viral life cycle. AMPs with antiviral activity became a research hotspot, with enormous promise to become available as antiviral drugs [50,51]. The antiviral mechanism may involve the inhibition of viruses by integrating into either the viral envelope or the host cell membrane (Figure 3B). This includes AMPs’ ability to integrate and damage viral envelopes, making the viruses unable to infect host cells; for instance, indolicidin antiviral activity has been related to its membrane-disruptive properties [52]. Likewise, most AMPs reduce the binding affinity between viruses and host cells; for instance, dermaseptin hinders the virus–cell interface [53]. Some AMPs occupy specific receptors on mammalian cells, and thus prevent viral particles from entering host cells; for instance, some α-helical cationic peptides bind to heparan sulfate, which is key to the attachment of HSV viral particles to the host cell surface, preventing the infection [54]. Interestingly, some AMPs can also cross the cell membrane, and once in the cytoplasm, can support the host’s defense system to fight against viruses or block viral gene expression, preventing cell-to-cell spread of viral particles [55]. Cecropin-A is active against HIV-1, HSV1 and 2, and the arenavirus Junin virus (JV) [56,57]. Melittin is active against HSV, HIV-1, and JV [57,58,59]. Alloferon 1 and 2, derived from the hemolymph of the blow fly *Calliphora vicina*, are active against influenza virus A and B [60]; alloferon 1 also showed activity against HSV-1 [61]. The six amino acid residues N-myristoylated-peptide, from larval hemolymph of the tobacco budworm *Heliothis virescens*, is active against HIV-1 and HSV-1 [62]. An homology modeling study demonstrated that temporins may provide therapeutic applications against MERS-CoV [63]. The antiviral activity of defensins has been widely demonstrated against several viruses, including HIV, influenza virus, human adenovirus, severe acute respiratory syndrome coronavirus (SARSC), papillomavirus (HPV), respiratory syncytial virus (RSV), and herpes simplex virus (HSV) [64]. Defensins likely block viral infection through several mechanisms; in fact, defensins have a direct action on virus particles, or interfere indirectly at various stages of the viral life cycle [65,66]; animals treated with rhesus theta-defensin 1 (RTD-1) showed a marked reduction in mortality in the presence of SARS-CoV [67]. Cathelicidins are linear peptides widely found in nature; humans express a cathelicidin known as LL-37, which is a 37 amino acid peptide with a Leu-Leu motif with antiviral activity against a number of viruses including HIV-1, IAV, RSV, rhinovirus (HRV), vaccinia virus (VACV), HSV, ZIKV, and hepatitis C virus (HCV), mediated primarily by its interaction with the outer envelope of a virus [65,68].

Some peptides, namely cell-penetrating peptides (CPPs), constituted by a short sequence of amino acids with a positive net charge and a high content of hydrophobic residues, are also able to enter cells, transporting cargoes of several dimensions [69,70]. CPPs transiently and locally disrupt membrane bilayers; when used in very low concentrations, they are likely carriers, while when used at higher concentrations, they show antimicrobial properties (Figure 3C). Thus, many CPPs exhibit cross-functionality with some AMPs, able to cross mammalian cell membranes by non-damaging processes and displaying considerable antimicrobial activity [71,72]. Some examples of peptides with double activity are SynB1, Pep-1-K, and gH625 [73,74,75].

Interestingly, some CPPs are now being used against biofilms. The peptide gH625 is effective in both inhibiting and eradicating several biofilms of single or mixed bacteria [76,77]. The peptide showed a low activity against planktonic cells, while it impaired formation of polymicrobial biofilms, influencing the biofilm architecture, and interfering with cell adhesion and polymeric matrix. The mechanism of action is not entirely known; nonetheless, gH625 does not form pores in the target membrane, but it acts on the biofilm structure, and this may explain its low activity against planktonic cells. Its action has been attributed to its capability to interact with model membranes, to penetrate the lipid bilayer, and to induce fusion between membranes [78,79,80]; additionally, the positive charges at the C-terminus may facilitate the interaction with negatively charged bacterial membranes and/or the matrix of the biofilm. The mechanism of action of gH625 support the hypothesis that this peptide could be exploited to facilitate the entrance of conventional drugs, promoting a synergistic activity that could also affect targets inside the cells, and opening up a wide range of possibilities, such as the use of CPPs as antibiofilm molecules in combination with conventional drugs [77]. Interestingly, gH625 was also evaluated for its antiviral activity (Figure 3D), and its infectivity inhibition was likely due to the formation of inactive aggregates between the fusogenic stretches present in the viral protein and the synthetic peptides. Aggregates formed as a consequence of intrinsic oligomerization, and likely stabilized a pre-fusion intermediate and prevented merging of the bilayers, indicating the potential role of CPPs as antiviral peptides [81].

Other peptides show antiviral activity that is correlated to the interference with essential mechanisms exploited by the virus to interact with the host cell, merge with it, and replicate within it (Figure 3E). Many viruses (enveloped viruses) are characterized by the presence of glycoproteins on their external surface. The merging of the viral and host-cell membranes represents a crucial step of the entry process, which is mediated by these glycoproteins [82]. During entry, viral glycoproteins go through a precise set of conformational rearrangements, which enable the close apposition of the two membranes, and ultimately produce a fusion pore, resulting in the release of the genetic material inside the host cell [83]. Antiviral peptides may inhibit virus attachment and fusion of host cells with viral membranes, disrupt the viral envelope, and inhibit viral replication. Peptides inhibit viral attachment and penetration, competing for receptor sites or interfering with the numerous conformational modifications of surface glycoproteins that are required for viral fusion. Entrance-blocking peptides are very promising candidates for viral therapy applications [84]. Most fusion proteins possess heptad repeat domains (HRs); HR domains are separated in the pre-hairpin intermediate, but following to the binding to cellular receptors, they fold one on the other, leading to the formation of a 6-helix bundle (6HB), and eventually resulting in fusion of the viral and host membranes [85]. Peptides are able to bind to their complementary HR region in the pre-hairpin intermediate, and preventing HR refolding into the stable 6HB structure mandatory for the fusion process may be exploited to inhibit virus entry. For instance, HSV peptide analogues to helical domains of the glycoproteins gH and gB inhibit viral infection [86,87,88]. In particular, sequences mirroring domains of gH or gB are appropriate to interfere with the fusion event, likely hindering the conformational rearrangements from pre- to post-fusion structures. One of these, derived from the gB glycoprotein, was also effectively conjugated to polyethylenglycol and cholesterol, and showed enhanced activity [88]. Antiviral peptides against the novel coronavirus represent an urgent challenge; a lipidic analogue of the HR domain of the SARS-CoV spike protein is an effective viral fusion inhibitor [88,89]. Peptides modeled from HR of several fusion proteins have been developed as a promising antiviral strategy; the most widely known is T20 (enfuvirtide), which is in clinical use for HIV-1 [90]. The most representative mechanisms are reported in Figure 3E.

### 2.2. The Use of Dendrimers

Dendrimers (Greek name derived from “Dendra” = tree) are interesting nanosystems characterized by a precise control over their structure, molecular weight, and number and location of functional groups over the dendrimer skeleton, which have been prepared and applied to several industrial and medical areas in the past 40 years [91]. Dendrimers are hyper-branched polymeric molecules endowed with an almost perfect geometrical three-dimensional architecture, with surface functionalities that provide them with unique properties completely different from those of linear polymers. Dendrimers have a central core to anchor the monomeric branches and create a polymer molecule; branches develop from the core into a globular structure, with a size of 2 to 5 nm, and with each section known as a generation (G1, G2, G3, etc.). The higher the generation, the more branches, and a higher number of exposed functional groups are available for conjugation with other molecules. There is also a direct link between generations and packaging of peripheral groups, and consequently presentation of cargos for both therapeutic and imaging applications. The coupling of drugs and targeting moieties are obviously correlated to the synthetic features of terminal functionalities. Those functionalities represent a perfect solution for coupling several copies of a drug and/or ligand to the periphery of the dendrimers. The arising interactions taking place between a dendrimer with multiple ligands on its surface and a target with several receptors point to a huge enhancement in the avidity between the dendrimer and the cell [92]. Hence, the presence of several ligands conjugated to a dendrimer scaffold can transform the structure into a high-affinity molecule.

Being a globular structure, nanodistricts can be created in the dendrimer that coexist, providing the dendrimer with different features between nucleus and periphery, and creating the possibility of encapsulation of active molecules. The dendrimer technology can also overcome some limits of pharmaceutical formulations, such as fast cellular entry, reduction of macrophage uptake, targetability, and easier crossing of biological barriers by transcytosis. In addition to their use as molecular carriers, dendrimers may present bioactive properties per se, and can also be used for material modifications to achieve nanohybrid materials. Cationic dendrimers have always attracted scientific interest because they are able to cross cell membranes by endocytosis, and some of them, unlike the cationic dendrimer completely functionalized with quaternary ammonium, are also able to escape the endosomal/lysosomal pathways through the proton-sponge effect and avoid premature destruction [93].

Dendrimers have been successfully exploited in medicinal chemistry. Their branched architecture appears to be essential to their application, and the plethora of ligands on their external shell has proved to be of major impact in inhibiting the multivalent adhesion activities between viruses, bacteria and cells, proteins, and combinations. Dendrimers can function as drugs themselves, or they can be carriers of a range of different drug molecules.

Several synthetic strategies that can be applied are summarized in Figure 4.

Various classes of dendrimers are able to inhibit microbial pathogens, and could find applications as antimicrobial agents, drug-delivery devices, or bacteriophobic coatings [94,95]. Cationic polyamidoamine (PAMAM) dendrimers, before being considered as active antimicrobials, were investigated as drug carriers to solubilize and deliver conventional antibiotics [96,97].

Dendrimer antimicrobial activity is strongly correlated to the effects of multivalency, deriving from the tree-like structure and clearly to the abundance of the active moieties. Cationic and/or amphiphilic dendrimers show antimicrobial activity associated with the disruption of the pathogen membrane [98,99].

The antibacterial activity depends on the electrostatic interaction between the positive charges of the dendrimer and the negative surface of the bacteria, on the progressive permeabilization of the bacterial membranes, and on the disruption of the lipid bilayer. Thus, similar to other antimicrobial materials, the multivalency in terms of positive charges plays a key role in their antimicrobial activity, and high-generation cationic dendrimers proved to be biocides with high activity [100]. Poly(amidoamine) (PAMAM) dendrimers are the most widely characterized, and have the strongest antibacterial activity. Generation 3 or higher PAMAM-NH_2_ dendrimers showed strong cytotoxicity, which impedes their further exploration as antibacterial compounds in vivo [101]. Chemical modifications of PAMAM-NH_2_ dendrimers reduced toxicities, but simultaneously decreased antibacterial activities [102]. Low-generation PAMAM-NH_2_ dendrimers present broad-spectrum antimicrobial activities, excellent therapeutic efficacy, and rather low cytotoxicity, and do not induce bacterial resistance [103].

A comparison between two first-generation cationic dendrimers, namely the amine terminated (G1-NH_3_^+^) and the free OH (G1-OH), showed their bactericidal activity on *Escherichia coli.* G1-OH showed lower bactericidal activity compared to the G1-NH_3_^+^, emphasizing the predominant role of cationic groups [104]. However, cationic dendrimers are toxic to mammalian cells, and in order to preserve the antibacterial effect with a significant reduction in term of toxicity on eukaryotic cells, nitric oxide (NO)-releasing dendrimers were developed [105,106]. The synergy of NO and cationic dendrimers allowed a decrease in the concentration of cationic dendrimers, while maintaining a satisfactory antibacterial activity and reducing the toxicity against mammalian cells (Figure 5A).

The modification of the PAMAM cationic amino groups with PEG chains showed a drastic reduction in toxicity problems without reducing antimicrobial capacity [107]. De Queiroz et al. conjugated dendritic polyglycerol (PGLD) with chitosan to achieve PGLD–chitosan dendrimers able to overcome bacterial proliferation of *Staphylococcus aureus* and *P. aeruginosa* [108].

Fourth-generation poly(propylene imine) dendrimers with their surface modified with maltose have been investigated for their antibacterial activity against the Gram-positive bacteria *S. aureus*, and *Staphylococcus epidermidis*; and against the Gram-negative bacteria *E. coli*, and *P. aeruginosa* and yeast *C. albicans*. The dendrimers showed the greatest antimicrobial activity against *S. aureus* [109].

Linear and branched poly(ethylene imine)s (L- and B-PEIs) were evaluated against *S. aureus* and *E. coli,* and exhibited enhanced activity against *S. aureus* [110]. The protonated ammonium groups of the PEIs are cationic, while the non-protonated amine groups and ethylene backbone are the hydrophobic moieties, which create the necessary cationic amphiphilic structures needed to induce membrane disruption without any further chemical modification of the surface groups.

Recently, poly(aryl ether) PAMAM-based amphiphilic dendrimers with different terminal spacers (containing amines, esters, and hydrazine units), and with a high propensity to self-assemble, were evaluated for their antibacterial activity against *E. coli* and *S. aureus* [111]. The results showed that amphiphilic dendrimers with an optimal surface-charge-to-hydrophobicity ratio displayed electrostatic interactions with the membrane, followed by insertion, effective membrane disruption, and bacterial death [111].

A new family of cationic carbosilane dendrimers containing imidazolium salts and their analogues with Ag(I)-NHC fragments in the periphery were evaluated for their antibacterial activity against three different bacterial strains, two Gram-positive (*S. aureus* and *Bacillus subtilis*) and one Gram-negative (*E. coli*). All compounds presented promising bactericidal activity [112].

A recent work investigated the role of different peripheral groups in the dendrimer penetration into *P. aeruginosa* biofilms [113]. The pH-responsive NH_3_^+^ dendrimers initially penetrated faster into the acidic environment of *P. aeruginosa* biofilms than did dendrimers with OH or COO^−^ groups at their periphery. In addition, the electrostatic attraction between the dendrimers with NH_3_^+^ peripheral groups and the negatively charged biofilm components allowed the dendrimer to accumulate near to the top of the biofilm. Nonetheless, accumulation of dendrimers with OH and COO^−^ peripheral groups was higher and more evenly distributed across the depth of the biofilms compared to NH_3_^+^ dendrimers. Thus, penetration and accumulation of dendrimers into biofilms is finely tuned by their surface composition, which is a crucial finding in the further development of new antimicrobial or antimicrobial-carrying polymers.

Dendrimers conjugated with anionic groups showed lower antimicrobial activities, although they have been exploited in combination with cationic dendrimers and in synergy with antibiotics to reduce the dose of the drug [96,114].

Furthermore, many dendrimer-based systems showed antiviral activities that either prevent binding of viruses to the target cell surface or prevent replication of the viral genome.

Carbohydrate binding proteins are often present on viruses and play a key role in invasion of the host cell where carbohydrates may act as receptors for viruses. Dendrimers functionalized with sialic acid with sizes similar to influenza virus successfully inhibited interactions with the virus [115]. Sialic acid functionalized dendrimers were able to inhibit hemagglutination of influenza virus more than monomeric sialic acid [116]. PAMAM dendrimers functionalized with sialic acid were effective against infection with murine influenza pneumonitis [117].

Poly-anionic dendrimers are able to inhibit HIV, HSV, and many other viruses that target the viral life cycle, and can prevent virus binding to the host cell [118]. Polyanionic carbosilane dendrimers showed in vitro and in vivo activity against HSV and HIV [119,120]. The polysulfated galactose derivatized poly(propyleneimine) dendrimers inhibited the infection of laboratory-isolated HIV-1 as efficiently as dextran sulfate (Figure 5B) [121].

Carbosilane dendrimers with sulfate groups at the periphery were shown to inhibit HIV-1 infection. In particular, the dendrimers did not cause the formation of gp120-CD4 (where CD4 is the receptor and gp120 the surface glycoprotein) complexes in correct numbers; as a consequence, they were able to prevent HIV-1 entry into the cells. The authors showed that although the dendrimer does not appear to directly block the CD4 binding site of gp120, the contact zone between both proteins was perturbed [122,123,124].

Other dendrimers conjugated to different anionic ligands have been exploited against HIV [121,125,126,127]. Similarly, most of them block gp120 [122,123], and thus viral penetration, but others, such as carboxylated fullerene-based dendrimers, are able to target later stages of the replication cycle [128]. Viologens, which are dendrimers based on N-alkylated 4,4-bipyridinium units, showed potent anti-HIV properties [127].

## 3. Peptide-Conjugated Dendrimers

To achieve a compromise between activity, biodegradability, selectivity for bacterial cells, and limited hemolytic toxicity and cytotoxicity towards eukaryotic cells, peptide dendrimers have been developed. The antimicrobial activity of dendrimers is intrinsically different from the mechanism of action of peptides; while the specific secondary structure of peptides is key to their activity, electrostatic interactions are the main driving force for dendrimers. Peptide dendrimers also emerged as promising nanosystems to overcome antimicrobial resistance [129]. This study focused on advantages in antimicrobial and antiviral therapy using nanosystems composed by dendrimers functionalized with peptides. The challenge is represented by the exploitation of both technologies to create a more soluble and less degradable nanosystem for the treatment of infectious diseases. 

### 3.1. Peptide-Conjugated Dendrimers with Antibacterial Activity

Peptide dendrimer structures exhibit several benefits over corresponding peptides or dendrimers alone. Peptide dendrimers are easily synthesized, preserve their activity at low- and high-salt conditions, resist proteases, are generally less cytotoxic, and act at lower concentrations. Several peptide dendrimers with antibacterial activity have been developed.

Natural cationic AMPs have stimulated the design of synthetic AMPs with repeating sequences of arginine and tryptophan [130]. The chemical features of positively charged arginines confer the possibility of mediating electrostatic interactions with the cell wall, while the lipophilic moieties of tryptophans interact with membranes, compromising their integrity [131]. The presence of these sequences on multivalent scaffolds further enhances the antibacterial activity [130]. These peptide dendrimers showed a higher microbial activity and lower hemolytic toxicity compared to the linear peptide and the dendrimer alone. Furthermore, extended exposure to sub-lethal doses of peptide dendrimers stimulated much lower levels of resistance compared to traditional antibiotics or antimicrobials in multidrug-resistant strains [130].

Three fifth-generation polyester-based dendrimers (G5Ds) decorated on their surface with amino acids were examined as potential antimicrobials against a huge selection of Gram-positive and Gram-negative human pathogens (*P. aeruginosa*, *S. maltophilia*, and *A. baumannii*). Peripheral hydroxyls groups were esterified with lysine (G5K), histidine (G5H), or a mixture 46/50 of histidine and lysine (G5HK). G5K proved to be even more active than the potent colistin against *P. aeruginosa* [132].

Peptide dendrimers based on di-, tetra- and octavalent lysine cores bound with tetra-(RLYR) or octapeptides (RLYRKVYG) elicited a strong antibacterial activity against both Gram-positives and Gram-negatives, with a major increase observed against Gram-negative bacteria [133]. The in vitro activity of the third-generation antimicrobial peptide dendrimer containing the dipeptide sequences KL (G3KL) showed significant activity against *A. baumannii* with minimal hemolytic toxicity [134,135]. A higher antimicrobial potency was correlated with a higher charge density and branching, and to a higher lipophilicity of the residues located at the C-terminus. G3KL presented good broad spectrum activity, with high selectivity for bacterial cells and low hemolytic properties [136]. A virtual screening performed on G3KL allowed to select a peptide-dendrimer (T7) as the most promising [137] with significant activity, combined with excellent stability in the presence of serum, and negligible hemolytic activity. Second-generation (G2) peptide dendrimers (TNS18) carrying a fatty-acid chain in the core powerfully kill Gram-negative bacteria such as *P. aeruginosa* and *A. baumannii*, and Gram-positive bacteria such as methicillin-resistant *S. aureus*. TNS18 likely adopts a hydrophobically collapsed conformation in aqueous solution with the fatty-acid chain backfolded onto the peptide dendrimer branches; upon interaction with the membrane, the dendrimer unfolds to expose its lipid chain and hydrophobic residues, thus enabling membrane disruption, and leading to rapid bacterial cell death [138].

A series of tryptophan-terminated dendrimers that were evaluated against clinical isolates of antibiotic resistant *E. coli* strains exhibited high potency, stability in human plasma, and very low hemo- and genotoxicity [139]. The outcome of multiple positive charges necessary for membrane disruption was equilibrated by the anchoring role of tryptophanes. Indeed, the tryptophan-terminated antimicrobial dendrimers, despite having the same net positive charge and branched scaffold construction as lysine-terminated molecules, showed a significantly higher selectivity and potency [139]. Amphiphilic Trp-rich peptide dendrimers were also proven effective against *Candida* cells that presented resistance against the commercially available antimycotics [140].

Scorciapino et al. [141] prepared the compound SB056 using a peptide, obtained by alternating hydrophilic and hydrophobic amino acids to achieve a membrane-active peptide able to form amphiphilic β-strands in a lipid environment. SB056 induced aggregation on the membrane surface and showed high activity against multidrug-resistant Gram-negative bacteria compared to that of colistin and polymyxin B. It also showed low levels of hemolytic toxicity, and it may represent a potential therapeutic alternative to conventional antibiotics to treat infections by multidrug-resistant bacteria.

A second-generation peptide dendrimer (2D-24), containing lysines and tryptophans, was investigated against planktonic, biofilm, and persister cells of *P. aeruginosa* [142]. The 2D-24 was able to penetrate the biofilm matrix; moreover, it proved to have a promising synergistic effect when administered in combinations with ciprofloxacin, tobramycin, or carbenicillin. Another peptide dendrimer was obtained by conjugation with three linear peptides containing arginines and tryptophans ((RW)n-NH_2_, where n is 2, 3, or 4) [143]. The bactericidal activity of (RW)_4_ dendrimer was tested against planktonic persister cells, planktonic regular cells, and persister cells in preformed biofilms of *E. coli*. When the bactericidal activity of (RW)_4_ dendrimer was evaluated on *E. coli* persister cells residing in 24 h sessile biofilms, the results suggested that all persister cells were eliminated.

An amphiphilic peptide dendrimer known as BALY was proven to be very effective against multi-resistant bacteria [144]. The activity of first- and second-generation cationic carbosilane dendrimers functionalized with a maleimide molecule on their focal point to allow thiol conjugation to three different AMPs was investigated against *S. aureus* and *E. coli*. One of these compounds (AMP3-dendrimer) was able to permeabilize bacterial membranes, causing significant morphological alterations and cellular-integrity damages. Moreover, antibacterial activity of the AMP3-dendrimer showed the same values as dendrimer alone against *E. coli* (MIC 16 mg/L); nevertheless, it is necessary to remark that the dendrimer concentration in the nanoconjugate was just 4.2 mg/L. This effect is pronounced against *S. aureus*; the reduction went from 128 mg/L in the free dendrimer to 16.9 mg/L in the AMP3-dendrimer [145].

Overall, peptide dendrimers present several advantages over corresponding antibacterial peptides. In fact, they are easily synthesized, resistant to proteases, are less cytotoxic, and function at lower concentrations.

The synthetic flexibility and high density of the peripheral groups of peptide dendrimers make them highly attractive as delivery systems to both kill bacteria and reduce toxicity against mammalian cells. Interestingly, peptide-based vaccines are promising candidates in the prevention of infectious diseases; it is thus necessary to preserve in vivo peptide stability and immunogenicity to achieve protective immunity. Recently, a dendrimer was developed as a peptide-based vaccine delivery system against *Chlamydia trachomatis* [146]. The carrier, which consists of a PAMAM dendrimer bound to Peptide 4 (AFPQFRSATLLL), was able to induce *Chlamydia*-specific serum antibodies after subcutaneous immunizations. The authors clearly demonstrated that dendrimers constitute a promising platform for the delivery of peptide vaccines. This could be exploited for other infectious intracellular bacteria and viruses, demonstrating the importance of delivering AMPs to the site of action.

### 3.2. Peptide-Conjugated Dendrimers with Antiviral Activity

Viruses, unlike bacteria, are more difficult to fight, due to their structure and infection capacity, but dendrimers can be used as vectors to fight viral infections. It is important to stress the difference in the mechanism of action between small-molecule antivirals and dendrimer antiviral drugs. The activity of dendrimer drugs is based mainly on interfering with the virus–host cell interactions, thus being correlated to the structural features of the polymer. Likewise, the high molecular weight and the presence of multiple sites for binding will lead to a steric hindrance hampering the virus–cell interactions by binding to either viral or cellular components [128].

A sulfonated poly-lysine dendrimer with a benhydryl amino core that was evaluated against intravaginal HSV and SIV/HIVchimera viruses was able to inhibit viral absorption, infection, and DNA synthesis [147,148,149].

VivaGel^®^ is the best microbicidal dendrimer known today. VivaGel^®^ is based on the dendrimer SPL7013 (Figure 6), which contains a polycationic surface responsible for virus binding and thus, for blocking virus binding/uptake by the cells, preventing infection. This anionic G4-poly(L-lysine)-type dendrimer with 32 naphthalene disulphonate groups on the surface showed potent topical vaginal microbicide activity, and prevents transmission of genital herpes (HSV-2), HIV, and other sexually transmitted infections (e.g., human papillomavirus). VivaGel^®^ condoms are already available in some countries, such as Japan, Australia, and Canada, and the company is engaged in studies against other important sexually transmitted viruses, such as the Zika virus [150].

Furthermore, dendrimers with peripheral tryptophan residues have the ability to inhibit an early phase of the HIV replication cycle through an interaction with envelope glycoproteins [151].

PAMAM-based dendrimers substituted with gH625, a membrane-interacting peptide derived from the surface glycoprotein H of HSV-1, are active against this important viral pathogen. Interestingly, the dendrimer itself was able to inhibit virus replication, albeit to a lesser extent compared to the peptidodendrimer, while the peptide on its own was unable to show any inhibition when tested at a concentration corresponding to the peptide present on the peptidodendrimer. gH625 was shown to significantly increase dendrimer penetration into the cellular matrix, mainly through a non-active translocation mechanism [152,153,154]. Currently, we are developing dendrimers conjugated to other interesting peptide candidates against HSV-1.

A family of carbosilane dendrimers homogeneously functionalized with a hemagglutinin-binding peptide (Ala-Arg-Leu-Pro-Arg) was evaluated against two different human influenza viruses (H1N1 and H3N2). The IC50 values of the best peptide dendrimer for both strains was 0.60 μM, while >100 μM for the peptide alone for both strains [155]. A dendrimer made of a polyglycerol scaffold covalently conjugated with PeB peptides, bearing high affinity for influenza A virus hemagglutinin, was proven to be effective in enhancing antiviral activity of peptides with no cytotoxicity. The authors demonstrated that by increasing the size of the dendrimer scaffold and adjusting the peptide density, they could reduce significantly viral infection [156].

## 4. Conclusions

Bacterial and viral infectious diseases remain a major threat to global health in the 21st century. Although research is aimed at developing new drugs, only a few new drugs have been licensed. Bacterial and viral infections are often difficult to address, and epidemic and pandemic events such as the novel coronavirus (COVID-19) emergency underscore the importance of developing new active drugs/vaccines.

Although many antibacterial and antiviral peptides have been developed and extensively studied, their clinical translation development has only achieved limited success. One of the major issues that prevents clinical applications is the discrepancy between in vitro and in vivo efficacy; essentially the clinical application is overcome by several major challenges, such as toxicity, low in vivo efficacy, poor enzyme stability, and high costs of production. To solve these problems, synthetic approaches have been attempted, and peptidomimetics have been developed with enhanced activity, safety, and stability. In this context, a key concept widely encountered in nature is multiplicity; in fact, branched structures are frequent in science and technology. Nature offers many examples of how dendritic structures are exploited in the animal and plant world; indeed, dendritic structures offer the possibility to expose a highly multivalent surface to ensure maximum interaction with surroundings. To incorporate dendrimers in biological agents, it is necessary to consider that the profile of a dendrimer is, to a large extent, regulated by the size of the dendrimer and the surface groups present on the individual dendrimer. Likewise, the core of the dendrimer is of less importance, as interaction takes place via the groups exposed on the surface, which may enable the dendrimer to penetrate. Peptide dendrimers represent an opportunity to exploit antimicrobial peptides and enhance their activities [157]. Table 1 shows examples of peptides, dendrimers, and peptide dendrimers for clinical applications.

Several peptides have been approved for clinical use; Table 1 shows several commercialized AMPs, as well as an antiviral peptide used against HIV-1. It is not surprising that most of them are designed for topical use only, which avoids unpredicted systemic toxicity, increases local effective concentration, and decreases the chances of degradation.

The only peptide dendrimer in clinical use is VivaGel^®^, which has been approved and marketed by Starpharma. It was marked for three applications: as a condom lubricant; as a mucoadhesive gel for treatment and prevention of BV (bacterial vaginosis); and as a gel for prevention of sexually transmitted infections (STIs), including HIV, HPV, and HSV-2. It is already commercially available as condom lubricant in Japan, Australia, and Canada under the brand LifeStyles^®^ DualProtect™. It has been approved in Australia and Europe for BV, and its approval in the USA and other countries is pending. It is still not commercially available for the prevention of STIs in any countries. Furthermore, it also shows significant activity against other viruses, such as the novel coronavirus SARS-CoV-2 [172], enabling a fast-track development of tools to fight COVID-19.

Our review focused on applications concerning peptide dendrimers and antimicrobial applications. There are several companies that are developing dendrimer- and peptide-based products mainly in the diagnostic field, which highlights the opportunities offered by dendritic materials in the prevention of and fight against many pathologies.

## Figures and Tables

**Figure 1 pharmaceutics-13-00101-f001:**
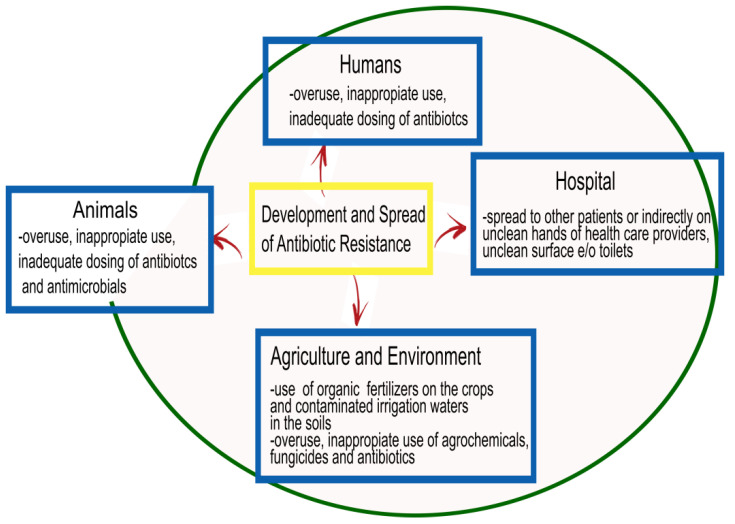
Factors involved in the diffusion of antibiotic resistance: human medicine in the community and in the hospital, animal production, and agriculture and environment.

**Figure 2 pharmaceutics-13-00101-f002:**
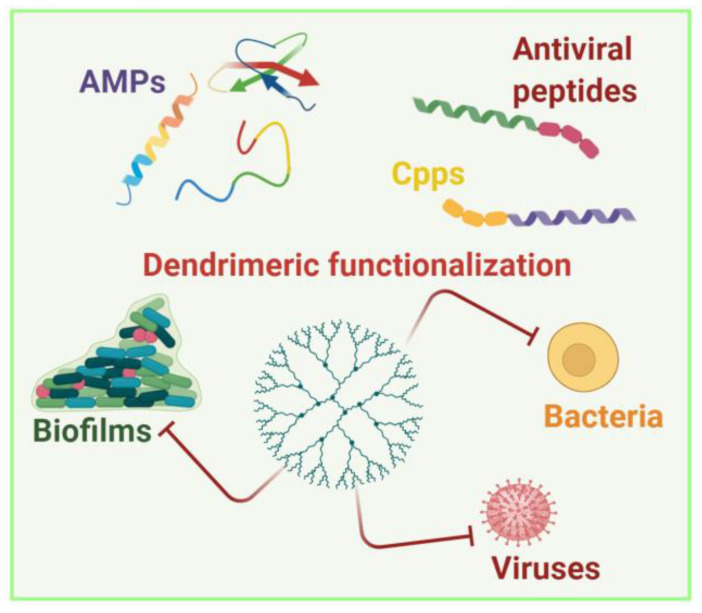
Dendrimers can be functionalized with antimicrobial peptides (AMPs), cell penetrating peptides (CPPs), and antiviral peptides for applications in fighting bacteria, viruses, and biofilms. Created with BioRender.com.

**Figure 3 pharmaceutics-13-00101-f003:**
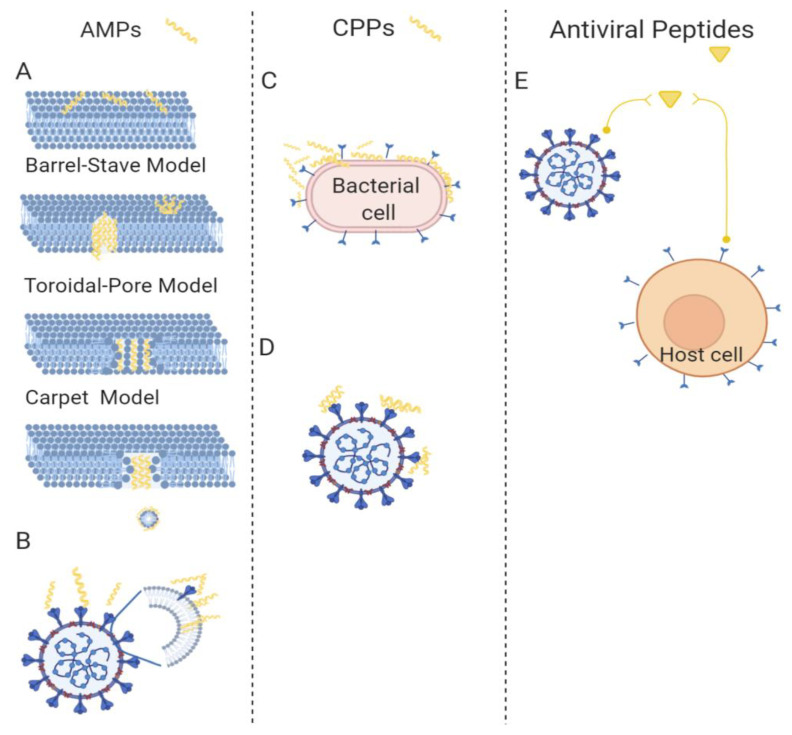
The main classes of peptides showing antibacterial and antiviral activity. (**A**) The mechanisms of antibacterial activity proposed for AMPs (barrel-stave, toroidal-pore, and carpet). (**B**) The hypothesized antiviral mechanisms for AMPs, which involve inhibition of viruses integrating in either the viral envelope or the host cell membrane. (**C**) The antibacterial mechanism proposed for CPPs, which involves local disruption of bacterial membrane bilayers. (**D**) The antiviral mechanism of CPPs, involving the formation of inactive aggregates between the fusogenic stretches present in the viral protein and the host cell. (**E**) The mechanism of antiviral peptides, involving inhibition of viral attachment and penetration. Created with BioRender.com.

**Figure 4 pharmaceutics-13-00101-f004:**
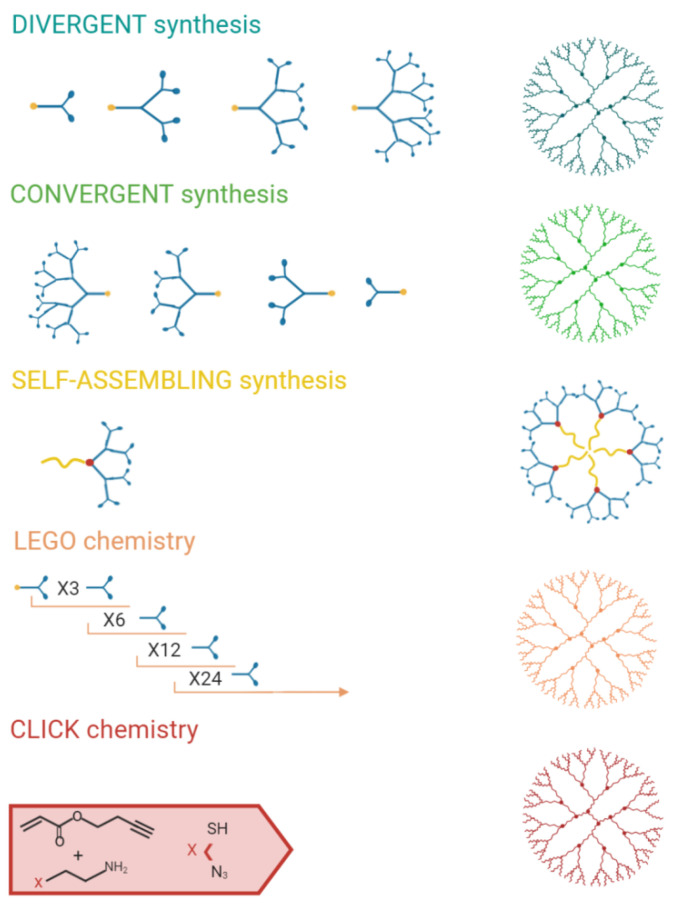
Synthetic strategies that can be exploited in dendrimer development. Created with BioRender.com.

**Figure 5 pharmaceutics-13-00101-f005:**
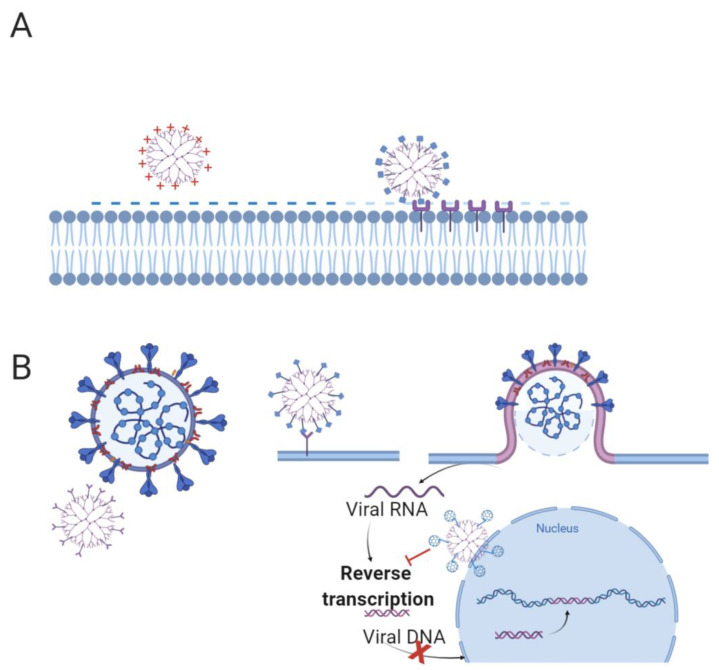
(**A**) Cationic dendrimers and maltose-conjugated dendrimers, the antimicrobial activity of which is related to the electrostatic interaction and consequent permeabilization of the bacterial membranes, or the interaction with specific receptors on the host-cell membrane. (**B**) Antiviral anionic dendrimers acting on different steps of viral infection. Created with BioRender.com.

**Figure 6 pharmaceutics-13-00101-f006:**
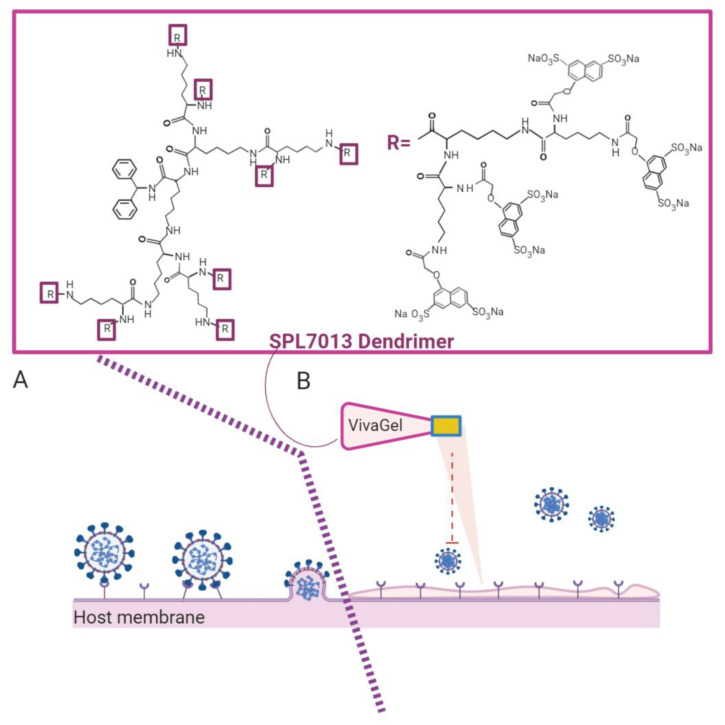
Representation of the SPL7013 dendrimer (VivaGel^®^) mechanism. (**A**) The host cell infected by the virus. (**B**) The host is protected from the infection by treatment with VivaGel^®^, which prevents the interaction and fusion of the virus with the host-cell membrane. Created with BioRender.com.

**Table 1 pharmaceutics-13-00101-t001:** Clinical applications of peptides, dendrimers, and peptide dendrimers.

Molecule	Characteristics	Application	Administration	Year of Approval	Ref.
**Peptides**					
Gramicidin D	α-helical peptide	Gram-positive bacteria: skin, eye, and wound infections	Topical	1940	[158,159]
Gramicidin S	Cyclic/β sheet	Gram-positive and Gram-negative: wound infections, spermicide, and genital ulcers	Topical	1942	[160,161]
Bacitracin	Cyclic peptide	Gram-positive bacteria: skin, eye, and wound infections	Topical	1948	[162]
Nisin	Lantibiotics	Gram-positive bacteria: food preservative		1960	[163]
Polymixin B	Cyclic lipopeptide	Gram-negative bacteria: meningitis, pneumonia, sepsis.	Parenteral, inhalation, intrathecal, and topical	1951	[164,165,166]
Colistin	Cyclic lipopeptide	Gram-negative bacteria: cystic fibrosis, intestinal infections	Topical, urinary tract, and inhalation	1959	[167,168,169]
Colistimethate (Prodrug of colistin)			Inhalation and parenteral formulation (Europe and Australia)		
Daptomycin	Cyclic lipopeptide	Gram-positive bacteria: endocarditis, skin infections	Topical and intravenous	2003	[170]
Enfuvirtide	Linear peptide	HIV-1	Subcutaneous injection	2003	[171]
**Peptide Dendrimer**					
VivaGel^®^	Poly-lysine	Bacterial vaginosis; condom for sexually transmitted infections	Topical	2012	[172,173]

## Data Availability

Data sharing not applicable.

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
