# Peer review of "Peptides and Dendrimers: How to Combat Viral and Bacterial Infections"

_pharmaceutics, 2021, doi:10.3390/pharmaceutics13010101_

Round 1

Reviewer 1 Report

The article by Falanga et al. is a short overview (including 164 references) dealing with the description of peptide dendrimers and their applications . Although not much is new, it might be a useful overview of this particular class of antimicrobials. Hence, the paper may be suitable for publication in Pharmaceutics, but it needs to be improved and updated.

Comments:

line 34: consider rephrasing "virus need to colonize a body to survive":

 virus need to colonize/invade a cell/living organism to reproduce.

Figure 1: antibiotics are also used in agriculture. Please, improve figure information (also in consistance with line 63). See, for instance: http://www.fao.org/antimicrobial-resistance/background/what-is-it/en/

line 130: referring to peptides: "...they have a low half-life, oral bioavailability is hampered by the intrinsic enzymatic action, polarity and molecular weight may limit intestinal permeability"

I understand "They" refer to linear, L-amino acid containing peptides. For instance, a cyclic peptide such as polymyxin (and also daptomycin, i. e.) have a long half-life (hours) and is used for selective digestive tract decontamination. Hence, they are quite enzymatically stable. A comment by the authors would be useful here.

Line 271: peptides analogue? analogues?

Line 437: “resist to proteases” à are resistant to proteases? resist proteases?

Line 548. “Corona virus”? better COVID-19, perhaps

Line 556 “…great importance deserves multiplicity” and “...multiply branched structures...” multiple? consider rephrasing

Line 563 likely or likewise? (also in lines 168, 204, 510)

Table 1: please, revise Update and correct:

Bacitracin cannot be considered a lipopeptide, it is a cyclic peptide.

Nisin is not active “per se” against Gram negative bacteria

Colistin is a cyclic lipopeptide. It should be mentioned that colistin is mostly used as colistimethate (CMS, prodrug, in Europe and Australia) for parenteral administration and by inhalation. Colistimethate was approved in 1959 (not 1970 as it is stated in the table). Colistin (in the sulfate form) is only used topically or orally for bowel decontamination. It is found in the injectable form only in a few countries (US, Brazil, Singapore and, Malaysia), like polymyin B.

See additional information, for instance in: https://www.unmc.edu/intmed/divisions/id/asp/protected-antimicrobials/colistin.html

Polymyxin B is used for parenteral (intravenous and intramuscular), aerosolized, intrathecal and, topical (ophtalmic use) administrations.

Polymyxin B was approved before 1970, I believe in 1951. Please, authors, check it. See for instance, https://www.nebraskamed.com/sites/default/files/documents/for-providers/asp/PolymyxinB_IV_Formulary%20(002).pdf

Polymyxin B is not generally used to treat urinary tract infections (UTI), due to low concentrations in urine. Colistimethate is the polymyxin derivative od choice.

Please, update information and references in the Table:

  1. Include for instance: Nat. Prod. Res., 2017,34, 886, https://pubs.rsc.org/ja/content/articlelanding/2017/np/c7np00023e#!divAbstract   https://www.tandfonline.com/doi/abs/10.1080/14712598.2017.1315402?tab=permissions&scroll=top

and others that authors may update.

  1. Reference 157 by Tsubery refers to the truncated polymyxin nonapeptide. It is not used in the clínic. It should be removed.
  2. Why reference 159 by Pines is cited? It refers mostly to treatments with carbenicillin and gentamicin. In addition, it also states that:” Colistin in our patients was useless and in some even deleterious, despite very high combined intramuscular and aerosol dosages. We cannot confirm others' success in apparently similar cases”. This reference should also be removed.

Author Response

Thanks to reviewer's comments, attacched our response

Reviewer 2 Report

As it follows from the title of the manuscript (“Peptide dendrimers…”), it should be devoted to peptides based nanosystems consisting of dendrimers functionalized with peptides, and focuses on their use to combat viral and bacterial infections. However, even from a formal point of view, it doesn’t correspond to these aims. The review consists of 24 pages, extracting 8 pages of References, we get 16 pages of the text, and only 3 of them describes Peptide conjugated dendrimers (chapter 3). To my mind, the manuscript requires significant corrections:

  • All chapters should be reduced, except of ch.3, that should be expanded;
  • Figures duplicate the text and should be refined;
  • To prove efficacy of dendrimers their activity should be compared with biological activity of linear peptides. It would be very desirable to introduce quantitative characteristics of activity (for ex., MIC and MBC for AMPs and AMPDs) and toxicity (for ex., HC50);
  • It would be important to explain improved activity (mechanism of action), decreased toxicity and sustained pharmacokinetic profile of peptide dendrimers, confirm it by experimental data and topology/sequence of the dendrimers;
  • Delivery of AMPDs to sites of action is important and interesting also;
  • AMPs have an influence on innate immunity and cytokine release. What’s about AMPDs?

Author Response

(The authors gave the same response as above.)

Reviewer 3 Report

It is well written, comprehensive, and quite informative review concerning peptide dendrimers and antimicrobial applications. It provides a relevant state of the art in this field. Therefore, in my opinion, the current manuscript meets the quality guidelines to be published in Pharmaceutics, after a few corrections below being addressed.

1. The image resolution of figures is not of publishable standard, authors should make it higher.

2. The comment of peylation would not induce immunogenic reaction in the host is not right (line 143-145). In recent studies, anti-peg antibodies had been found, which points pegylation may induce immunogenic reaction in the body.[1, 2] I strongly recommend the sentence of line 143-145 should be changed to ‘Covalent binding of hydrophilic polymers, including polyethylene glycol (PEG), zwitterionic polymers, and others,[3-5] produces better solubility, stability, and greater half-life in vivo.’ Some recent studies[3-5] should be cited.

[1] J. J. Verhoef, J. F. Carpenter, T. J. Anchordoquy, H. Schellekens, Drug discovery today 2014, 19, 1945.
[2] P. Zhang, F. Sun, S. Liu, S. Jiang, Journal of Controlled Release 2016, 244, 184.
[3] B. Børresen, J. R. Henriksen, G. Clergeaud, J. S. Jørgensen, F. Melander, D. R. Elema, J. Szebeni, S. A. Engelholm, A. T. Kristensen, A. Kjær, ACS nano 2018, 12, 11386.
[4] W. Lin, G. Ma, F. Ji, J. Zhang, L. Wang, H. Sun, S. Chen, Journal of Materials Chemistry B 2015, 3, 440.
[5] H. Du, F. A. de Oliveira, L. J. Albuquerque, G. Tresset, E. Pavlova, C. c. Huin, P. Guégan, F. C. Giacomelli, Langmuir 2020, 36, 1266.

3. Line 317-320, the sentence should be revised to make it more accurate. Not all the cationic dendrimers are able to escape from endosomal/lysosomal pathways via the proton-sponge effect, for example, fully quaternary ammonium functionalized dendrimer, one kind of cationic dendrimer, doesn’t have any proton-sponge effect.

Author Response

(The authors gave the same response as above.)

Round 2

Reviewer 1 Report

The manuscript is now acceptable for publication

Reviewer 2 Report

Peptides and dendrimers containing peptides are two different classes of biologically active molecules, and it would be of especial interest to compare their three-dimensional structure, functionalities, etc., and try to explain why dendrimers (sometimes) have higher efficacy, selectivity, and safety than linear molecules. However, after changing the title of the manuscript it can be accepted for publication.